# Factors at Insemination and Subsequent Conception of Cattle under Heat-Stress Tie-Stall Environments

**DOI:** 10.3390/ani14121763

**Published:** 2024-06-11

**Authors:** Siriporn Kanwichai, Duanghathai Saipinta, Sasithorn Panasophonkul, Witaya Suriyasathaporn

**Affiliations:** 1Faculty of Veterinary Medicine, Chiang Mai University, Chiang Mai 50100, Thailand; kanwichai.s@gmail.com (S.K.); panasasithorn@hotmail.com (S.P.); 2Overseas Division Cambodia Office, Asian Satellite Campus Institute, Nagoya University, Nagoya 484-8601, Japan; suriyasathaporn.witaya.y3@f.mail.nagoya-u.ac.jp

**Keywords:** secondary estrus signs, cervical mucus, conception risk, tie-stall housing

## Abstract

**Simple Summary:**

Heat stress, a significant reproductive performance problem for dairy cattle around the world with different mechanisms in each part of the world, is further exacerbated for cattle in tie-stall barns in the tropics due to the challenge of estrus detection. Our research, which we believe to be of significant importance, aimed to determine factors improving conception rates, including secondary estrus signs, reproductive tract characteristics, body temperature, and ambient temperature, during the insemination of cattle in the specified environment. We found that the best rates of conception were achieved when cattle were bred 24 h after initial secondary signs of estrus detection. Sticky cervical mucus during insemination increased conception risk, while elevated body temperature and increased ambient relative humidity decreased it. These findings, which we consider to be crucial, are instrumental in increasing conception rates for cows housed in tie-stall barns, providing valuable insights for dairy farmers in tropical environments.

**Abstract:**

This research aimed to compare the conception rates among AI at 12, 24, and 36 h after estrus in cattle living in tie-stall barns in a tropical environment. The second study was to determine factors at insemination at 24 h after estrus, including secondary signs of estrus behavior, reproductive tract characteristics, and heat-stress factors, in relation to conception. The study was conducted on 22 tie-stall dairy farms in Chiang Mai, Thailand. After secondary signs of estrus were observed, all farmers were informed for data collection at the insemination time. Repeated logistic regression models were used to determine factors associated with conception. The results revealed that cattle inseminated 24 h after estrus detection had the highest conception rate (67.5%). The study also found that cattle with three or two secondary estrus signs before insemination had a higher risk of conception than those with only one sign. Interestingly, there was an increased risk of conception when sticky or no cervical mucus was observed during insemination (OR = 6.85 and OR = 5.96, respectively). Moreover, increases in body temperature and ambient relative humidity were related to a decrease in the chances of conception. This study recommends delaying insemination to 24 h after the initiation of secondary estrus signs. Multiple signs of estrus, sticky cervical mucus, and lower body temperature increased conception risk in dairy cattle living in tie-stall barns in a heat-stress environment.

## 1. Introduction

The productivity and efficacy of dairy cattle heavily depend on their reproductive performance within a particular environment. In order to determine the most favorable moment for artificial insemination (AI), accurate estrous detection is crucial, given the influence of environmental factors on estrus behaviors [1]. Tie-stall barns are the primary type of lodging for dairy cattle in various countries such as the United States [2], Canada [3], Norway [4], and Thailand, where smallholder dairy farms also house their animals in tie-stall stables. Although there is minimal competition for feed in these systems, the tie-stall, even with a sufficient length of tied rope, inhibits the ability of heifers and cows to engage in typical social behaviors [5,6], including rubbing, head butting, licking, and mounting, which are crucial indicators of estrous expression in tie-stall confinement [7,8]. Both mounting and standing heat of the estrus cows are performed accompanied by the only adjacent cows that are mostly not in estrus. Estrus detection in tie-stalls is achieved through a combination of secondary indicators such as mounting, restlessness, bellowing, and vaginal discharge [6,9,10]. However, when determining the optimal time for artificial insemination, the suggested insemination time of 12 h following the initial observation of standing estrus [11] should be deferred due to these secondary estrus symptoms.

Globally, lactating dairy cows face a significant threat from heat stress, which can interfere with their reproductive processes and significantly reduce conception rates [12]. During periods of thermal stress, the survival of the fetus and the productivity of lactating dairy cows can decline by 30 to 60 percent [13,14] compared to non-thermal-stressed periods. Additionally, the incubation temperature of oocytes obtained from dairy cattle during mild months decreases continuously by 1 °C within the range of 38.5 to 40.5 °C, which can further impede reproductive processes [15]. Heat stress can also modify the dominance of the first phase of follicular growth, leading to compromised follicle development [16]. By altering the secretion of reproductive hormones, including follicle-stimulating hormone (FSH), luteinizing hormone (LH), estradiol (E2), and progesterone (P4), via the hypothalamic–hypophyseal–ovarian axis, heat stress can delay ovulation and reduce estrus indicators, ultimately leading to a reduction in the development of the dominant follicle and conception rates [17].

Blood biochemical profiles, the parameters of nutritional status, are beneficial for diagnosing metabolic disorders that subsequently relate to infertility in animals [17]. In heat-stressed dairy cows, a reduction in dry matter intake and, subsequently, negative energy balance causes metabolic disorders, e.g., decreased plasma concentrations of insulin, glucose, and IGF-I and increased plasma concentrations of GH and non-esterified fatty acid [17]. Both heat stress itself and negative energy balance affected pre- and post-ovulatory reproductive performance [12,17]. A scientific review refers to an in vitro study that showed preimplantation embryos are the most sensitive to elevated temperatures in a two-cell stage [12], indicating that heat stress and metabolic profiles during the insemination time are important for successful conception. However, no study, especially for tie-stall barns in the tropics, has identified the effects of heat stress.

In Thailand, most experienced farmers inseminate their cattle 24 h after seeing the secondary signs of estrus to cope with infertility under heat stress. Still, no study has been reported on the successful conception of these heat-stressed cattle in this condition in the tropics. To improve the reproductive performance of heat-stressed cattle in tie-stall barns, it is important to identify the secondary estrus signs, reproductive tract characteristics, or their combination 24 h after seeing the secondary signs of estrus that could enhance conception rates. Therefore, this study aimed to determine whether AI at 24 h after seeing the estrus signs had the best conception rates in cattle in tie-stall barns in the tropics (Study 1). Consequently, Study 2 aimed to identify the secondary estrus signs, reproductive tract characteristics, and their combination that could enhance the conception of heat-stressed cattle housed in tie-stall barns and inseminated 24 h after initial estrus.

## 2. Materials and Methods

### 2.1. Farm Selection and Management

Twenty-two small-holder dairy farms in Chiang Mai, Thailand, participated in this research. Each of these ranches operated for more than ten years and utilized tie-stall housing. The herds on these farms consisted of 20 to 30 crossbred Holstein–Friesian cows and 3–5 heifers. The tie-stall barns were rectangular and had concrete floors with no supplemental cooling systems. Some of the stalls had individual rubber mats, but most of them did not. At all times, the cows and heifers were secured with cords measuring between 1.5 and 2.5 m in length, depending on the experience of farmers and the farm environment, which permitted them to lie down and move. The feeding management strategy was determined by the roughage availability on each property and the milk production of each cow. In addition to ad libitum roughages, including grass, maize, dry corn, rice straw, corn silage, and bean hay, the cows were given commercial concentrates in proportion to their milk production. Feeding frequency and concentration and forage feed varieties varied between cows. The cows were milked twice a day using a bucket-type milking machine. Reproductive management included the monitoring of estrus in cows starting from 60 days postpartum and in heifers starting from 18 months of age. Estrous behavior was observed 2 times a day, and the times of observation were based on the farmers’ experience and availability, but mostly after milking for at least 30 min. Most artificial insemination procedures were conducted between 12 and 36 h following the initial symptom of estrus. The implementation of AI was dependent upon the experience of each farmer, as determined by estrus appearances and successful conceptions on their farm.

### 2.2. Animal Selection and Study Design

The study was conducted between February and July 2017, covering cool (February), summer (March–April), and rainy (May–July) seasons, in accordance with the Thai Meteorological Department’s declaration. The average ambient temperature and humidity in the cool season were 25.8 °C and 40–50%; in summer, they were 33.4 °C and 65–70%; and in the rainy season, they were 30.3 °C and 75–81%, respectively. Except for the inseminator, the participating farmers maintained the same management approach for their cows for the duration of the study. The researcher (S.K.) examined the reproductive tract and was the only inseminator for all AI procedures in this study. All healthy heifers at the age of 18 months and healthy lactating cows beginning 60 days postpartum were initially enrolled in the study. The study design is outlined in detail in Figure 1. After the estrus was detected, the farmers determined the voluntary period before AI and scheduled the researcher. The data recorded included estrus signs and the time elapsed from the initial observation of estrus signs to AI. All participating farmers were trained to define estrus behaviors (with or without) as bellowing, restlessness, vulva swelling and redness, vaginal discharge, mounting, and standing heat. The researcher collected management data, blood samples, and reproductive tract data using rectal palpation. The management data comprised body condition score (BCS), AI date data, parity, body temperature, and ambient temperature and relative humidity (RH). The ambient temperature and RH were measured by the researcher with a dry–wet bulb thermometer at one meter to the animal. The BCS was quantified on a scale of 1 to 5, with 5 indicating extreme fatness and 1 indicating extreme thinness [18]. In this study, the temperature humidity index (THI) was calculated using the formula described by Dikmen et al. [19], THI = (1.8 × T + 32) − [(0.55 − 0.0055 × RH) × (1.8 × T − 26)], where T represents the temperature in Celsius and RH means the relative humidity in percentage. Blood samples were obtained from the coccygeal veins of the animals and preserved in containers containing sodium heparin or EDTA (BD Vacutainer^®^, BD, Plymouth, UK) before the application of AI. The specimens were immediately chilled on ice before being transported to the Laboratory of the Chiang Mai University Faculty of Veterinary Medicine for examination. The reproductive tract characteristics of the animals were assessed at the time of insemination by the investigator in accordance with the established protocol during rectal palpation, as illustrated in Figure 1. The characteristics included vulva swelling; pink vulva membrane (pink or pale); uterine tone as mild (flaccid) or strong (turgid); and cervical mucus as none, watery, and sticky (stretchy and mucous-like). The animals were subsequently inseminated. Before AI, the frozen semen was thawed in 37 °C water for 30 s before being inserted into an AI gun. Semen was deposited into the uterine body using the recto-vaginal palpation method. Forty days after AI, pregnancy was diagnosed via rectal palpation; conception status was classified as either 1 for conception or 0 for non-conception. The calculation for the conception rate involved dividing the count of pregnant cattle by the count of inseminated cattle.

### 2.3. Blood and Plasma Chemicals, Metabolites, and Reproductive Hormone Measurement

The plasma was separated immediately upon delivery to the laboratory via centrifugation at 3000× *g* for ten minutes before being stored at −20 °C in a sterile 1.5 mL tube. The plasma from the EDTA blood sample was analyzed using an automated hematology analyzer (ABX Micros ESV 60, Horiba Medical, Kyoto, Japan) to determine the packed cell volume (PCV), red blood cell (RBC), hemoglobin (Hb), white blood cell (WBC), and platelet count. Meanwhile, the plasma from the heparin sample was analyzed for beta-hydroxybutyric acid (BHB) using the enzymatic method (Ranbut kit, Randox Laboratories Ltd., Crumlin, UK); non-esterified fatty acid (NEFA) using the colorimetric method (FA115 kit, Randox Laboratories Ltd., Crumlin, UK); and BUN, creatinine, alkaline phosphatase (ALP), and alkaline aminotransferase (ALT) using an automated chemistry analyzer (ABX Pentra 400, Horiba Medical, Kisshoin, Japan). Plasma levels of progesterone (P4) and estradiol (E2) were measured by enzyme immunoassay (EIA) following standard procedures [20].

### 2.4. Statistical Analysis

A summary of data was created for the number of AI services, the number of animals that were observed in standing estrus, conception rates, and conception rates by season across all the farms. To compare the conception risk among times to AI (Study 1), repeated logistic regression was used (Proc Genmod, SAS University Edition), which initially analyzed the data using all the data or data from each season. Time to AI was considered a fixed effect, while farm was considered a random effect. 

In Study 2, data from cattle inseminated at 12 and 36 h after having estrus signs were excluded from the final data set. When the number of primary estrus signs or standing estrus was less than 10%, data with standing estrus were excluded from the data set. Any estrus signs, reproductive tract characteristics, and combinations with less than 10% of total AI were not included in the final data set used for statistical analysis. The estrus signs and their combinations or reproductive tract characteristics and their combination were determined based on their natural occurrence, and the pairwise combinations were treated as separate parameters. Regardless of estrus signs or reproductive tract characteristics, the number of estrus sign combinations and reproductive tract characteristic combinations were defined as 1, 2, and 3 combinations. The conception rate was calculated as the number of pregnant cattle divided by the number of inseminated cattle. 

Before statistical analysis, the final data were tested for normality. Data on concentrations of E2, P4, ALP, BHB, BUN, and NEFA were not normally distributed, and log transformations were applied before statistical analysis. Continuous data included body temperature, ambient temperature, RH, THI, concentrations of plasma hematology, reproductive hormones, enzymes, and metabolites. In contrast, categorical data included season, parity, BCS, estrus signs, and reproductive tract characteristics at AI. Estrus signs, reproductive tract characteristics, heat-stress factors, and blood chemistry and reproductive hormones were defined as independent variables for determining factors related to successful conception. Due to the confounding of reproductive management within an individual farm, univariate analysis using repeated logistic regression models (Proc Genmod, SAS University Edition) was performed using the farm as a random effect, and each independent variable was tested as a fixed effect. The dependent variable was conception status, as 0 and 1. Parity, heat-stress measurement, individual and combined signs of estrus, individual and combined reproductive tract characteristics, and blood chemistry and reproductive hormones were included as independent variables. A final model was created using multivariate repeated logistic regression models, Proc Genmod (SAS University Edition), using the free entering method. The variables with *p* < 0.20 from the univariate analyses were used for inclusion in the final model analysis. The independent variable with the lowest *p*-value and *p* < 0.1 was serially added to the final model. The odds ratio, including lower and upper limits, was calculated for each independent variable.

## 3. Results

For the result from Study 1, the proportion of producers who opted to artificially inseminate their cattle 24 h after the detection of estrus (67.5%) was nearly four times that of the number of cattle inseminated at 12 and 36 h after the detection of estrus (16.25% each, Table 1). The conception rate of cattle inseminated at 24 h after estrus was the highest conception rate (53.7%) and there was a higher risk of conception than either 12 h (*p* < 0.05) or 36 h (OR = 0.39 and 0.44, respectively). Only five or 3.1% of the cattle included in the study were reported to be in standing estrus. The conception rate during the cool season was the highest (69.4%), whereas the lowest conception rate was observed during the rainy season (26.9%). 

For Study 2, a total of 108 observations from cattle inseminated at 24 h after estrus detection were included in the final data set. The environmental conditions during the study are shown in Figure 2. Ambient temperature, relative humidity, and THI increased substantially from the cool season to the rainy season. The THI during the cool season was typically lower than 72, but the THI was higher than 75 during the rainy season. The mean and SEM ambient temperatures during the cool, summer, and rainy seasons of this study were 23.2 ± 0.59, 25.8 ± 0.50, and 26.9 ± 0.34, respectively. The highest average RH and SEM was 73.8 ± 2.31% during the rainy season, whereas the RH during the cool and summer season was 52.3 ± 2.89 and 53.0 ± 2.16, respectively. The average THIs during the cool, summer, and rainy seasons were 69.2 ± 0.57, 72.9 ± 0.57, and 77.0 ± 0.35, respectively. The average body temperature and standard deviation of all cattle at the semination time was 38.57 ± 0.42 °C, ranging from 37.8 to 39.7 °C.

Cattle observed in standing estrus (n = 3) were excluded due to the small sample size. Therefore, the total sample size was 105. Most farmers used vaginal discharge (n = 75) and mounting (n = 62) as signs of estrus. By rectal palpation, all cattle were determined to have a uterine tone at AI, and only six and two cattle expressed pink vulva membranes and vulva swelling at AI, respectively. Therefore, these characteristics were excluded from the statistical analysis. The frequency of estrus signs and their combination, as well as reproductive tract characteristics and their combination according to conception results, are shown in Figure 3a,b, respectively. The most frequent combination of estrus signs was vaginal discharge and bellowing (n = 15); mounting (n = 12); or the combination of mounting, vaginal discharge, and restlessness (n = 12). Cattle expressing at least three estrus signs had a higher conception rate (72%) compared to those with only two (57%) or one sign (22%). The most common reproductive tract characteristics observed were sticky cervical mucus and its combination (n = 59) and no discharge and its combination (n = 30). Cattle with no, sticky, and watery cervical mucus had a conception rate of 36.7%, 69.5%, and 25%, respectively.

The univariate models of estrus signs and factors measured at AI in association with conception rate with *p* < 0.2 are shown in Table 2. Heifers tended to have a higher (*p* = 0.08) risk of conception (OR = 11.65) than cows at parity 1. Cattle inseminated during the cool season had a greater (*p* = 0.01) risk of conception (OR = 4.92) compared to those bred during the rainy season. Higher body temperature (*p* = 0.02) or RH (*p* = 0.01) in cattle at AI decreased the conception risk (OR = 0.07 and 0.93, respectively). The risk of conception for cattle that bellowed, had vaginal discharge, or had vaginal discharge and bellowed was approximately 2 to 3 times higher than cattle without those signs. Regardless of the estrus sign, cattle with three (OR = 9.2) and two (OR = 4.6) estrus signs had higher conception risks (*p* < 0.0001) compared with those only displaying one estrus sign. During AI, cattle with sticky cervical mucus had a higher risk (*p* < 0.001) of conception compared to those without any cervical mucus. The increased concentration of hemoglobin and estradiol at AI increased conception risk (Table 2). 

The final multivariate model of estrus and factors measured at AI on conception risks is presented in Table 3. Heifers had an 8.58 times higher conception risk than a cow in parity 1. Higher body temperature and RH were associated with decreased pregnancy risks (OR = 0.19 and OR = 0.93, respectively). Cattle with three (OR = 23.7) and two (OR = 12.8) observed estrus signs had a greater risk of higher conception compared with cows exhibiting only one estrus sign. When included with other factors, cattle with either sticky (OR = 6.8) or no (OR = 5.96) cervical mucus had higher conception risks than watery cervical mucus.

## 4. Discussion

Originally reported by Trimberger [11], the highest conception rates are obtained by postponing AI for 12 h following the observation of standing estrus. This recommendation, also known as the AM/PM rule, has been adopted globally [21] and is supported by the principal organizations for dairy improvement in Thailand, the Department of Livestock Development and Dairy Promotion Organization, and the Ministry of Agriculture and Cooperation. In our current blind study, producers did not have any information on the results of delaying AI for 24 h after initial estrus detection during the study. The findings from our experimental trial demonstrate that increasing the voluntary interval between initial estrus detection and AI to 24 h after secondary signs like bellowing or vaginal discharge or other signs shown in Table 2 are detected from the recommended 12 h for cattle houses in tie-stall stables under heat-stress conditions led to a higher conception rate (67.5%) (Table 1). To our knowledge, this is the first report on the use of a 24 h interval between the detection of estrus and AI. Changing the insemination interval from 12 to 24 h might have initially resulted in imitating farmers who had better conception rates in their herd compared to following recommendations. According to the results of our research, an AI interval of 24 h (53.7%) resulted in a higher conception rate than AI intervals of 12 or 36 h (38.5%) or 42.3% (Table 1). Proestrus in cattle is the period when progesterone declines as the CL regresses and estrogen concentrations increase, causing secondary signs of estrus to occur. Using multiple secondary signs of estrus as part of estrus detection in cows housed in a tie-stall barn instead of standing estrus improved conception rates when inseminating at 24 h after the detection of estrus in this study. 

When utilizing estrus detection to determine when to breed cattle, dairy producers often rely on secondary signs of estrus as additional indicators or to closely monitor the cow for standing estrus behavior. To determine whether cows exhibiting secondary signs will stand to be mounted, farmers may isolate the cow with a sexually active cow or teaser animal on non-slippery ground. Despite the cows in this research being restrained in their stalls with 1.5 to 2.5 m of rope, mounting appeared as the second most frequently reported indication of estrus (n = 62) after vaginal mucous discharge (n = 75), which was the most frequently reported sign. The results presented in Figure 2 indicate that cattle exhibiting three signs of estrus had a 72% conception rate, which was higher than the conception rates of cattle displaying two (57%) or one (22%) signs of estrus. The scoring system, including a combination of estrus behaviors, has been used to design the optimal time to AI [6,22], including mounting as the most frequently scored secondary estrus sign. In contrast, our results indicated that mounting alone was not related to the conception rate. However, the risk of conception was approximately doubled when bellowing and vaginal discharge were present (Table 2). When mounting was observed along with bellowing and vaginal discharge, the risk of conception increased. As the number of observed signs of estrus increased in both univariate (Table 2) and multivariate (Table 3) analyses, the risk of conception increased. Regardless of the estrus signs expressed, cattle with three estrus signs had the highest conception risk compared to cattle with only a single estrus sign. Observing more secondary estrus signs might provide more confidence to the farmer that estrus has initiated. The combination of estrous signs expressed, including interaction with neighboring cows or herders, vaginal mucus discharge, mounting, and standing to be mounted, are frequently used when a scoring system is used to assess the time to AI in a tie-stall system [23,24,25]. According to Yoshida and Nakao [25], the onset of secondary estrus indicators in cattle occurred 9.6 h before the maximum of standing estrus on average. Therefore, the highest conception risks using the secondary estrus signs and AI approximately 24 h after the detection of estrus would be similar to those of breeding 12 h after standing estrus.

When examining the reproductive tract at AI, cervical mucous, particularly viscous discharge, was the most observed characteristic (Figure 2). As shown in Table 2, the univariate analysis revealed a significant association between sticky cervical mucous and an increased risk of conception. However, the conception risk with both sticky (OR = 6.85) and no cervical mucus (OR = 5.96) after correction for other parameters in the multivariate model was not different. Still, it was higher than that with watery cervical mucus (Table 3). Both heat-stress parameters, body temperature, and environmental relative humidity were included in the multivariate model, which could explain why no cervical mucus was associated with an increased risk of conception. Contrary to findings from other research, conception rates were found to be highest in cases where cervical mucus was watery and occurred in the middle of estrus expression [26,27,28]. Cervical mucus is involved with sperm selection and transport, being the first medium spermatozoa must go through when ascending to the site of fertilization, and it is consequently favorable to sperm penetration [29]. During the progression of estrus, cervical mucus changes from none to sticky quickly and reaches the highest watery values at the middle of estrus, and then decreases rapidly until no discharge is observed at the end of estrus [30], which corresponds with the time of ovulation [26]. Using secondary estrus signs, however, the characteristics of no and sticky cervical mucus might have occurred either before or after the middle of estrus. Heat stress decreases follicular growth and reduces the length and intensity of estrus [31]. Follicular growth dynamics in cows are altered by heat stress, resulting in the longer duration of dominance of the preovulatory follicle due to its early emergence [16]. Therefore, the appearance of no or sticky cervical mucus 24 h after the secondary estrus sign might occur before the middle of estrus. Therefore, an increase in the duration of preovulatory follicle dominance might explain the higher risk of conception observed in cows lacking or having sticky cervical mucus. All cows in this study exhibited mild to strong uterine tone; consequently, the absence of uterine tone associated with a lower pregnancy could not be assessed in this study.

Both body temperature and relative humidity were included in the final multivariate model and are directly related to heat stress. In general, the cattle body temperature ranges from 38.0 to 39.3 °C during thermoneutral conditions. A higher body temperature and environment temperature on the day of insemination are strongly correlated with decreased pregnancy rates [32,33]. An increase in body temperature one day before insemination decreased the conception rate of lactating dairy cows [33]. In tropical environments, the lowest pregnancy rate observed occurs during the rainy season [34] due to the combination of high ambient temperature and high RH, which negatively affects embryo development and quality, increasing the failure of implantation. Furthermore, thermal stress compromises the uterine environment, which subsequently increases the loss of embryos [17].

Hemoglobin, hematocrit, alkaline phosphatase, creatinine, and estradiol measured at AI were positively related to conception risk in the univariate analysis, but they were not included in the final multivariate model. These blood parameters are related to health status, heat stress, estrus expression, and conception [23]; however, they were not included in the final model, possibly because they correlated with other factors in the final model. In the present study, heifers had a higher risk of conception compared to lactating cows. This finding is in agreement with previous studies [35,36,37], which reported lower pregnancy rates with the increased parity of cows. Disturbances in reproduction related to the development of a dominant follicle [38], hormonal stasis [38,39], and uterine environment [40], together with metabolic disturbances, including negative energy balance [37], contribute to the lower conception rate of older cows compared to heifers.

## 5. Conclusions

For cows housed in tie-stall barns in a tropical environment characterized by heat stress, delaying AI until 24 h after the detection of secondary estrus signs resulted in higher conception rates. Multiple signs of estrus, especially bellowing, mounting, and cervical mucus, could be used by producers to increase conception risks. Sticky or no cervical mucus increases the risk of conception. In contrast, relative humidity and body temperature on the day of insemination reduced the risk of conception. In contrast, THI and ambient temperature were not included as factors under the conditions of this study.

## Figures and Tables

**Figure 1 animals-14-01763-f001:**
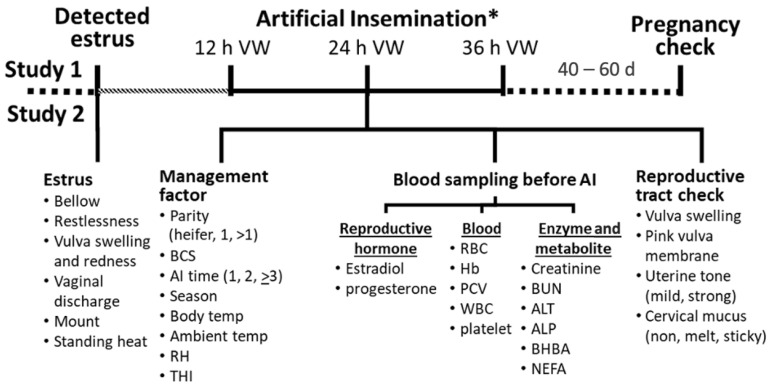
The study plan scheme, data, and blood collection of this study, as well as the estrus detection and 12 to 36 h voluntary waiting period (VW), were performed by farmers as routine procedures. Study 1 was to determine conception rates of AI at 12, 24, and 36 h VW and Study 2 was to determine factors related to conception after AI at 24 VW. Blood samples were collected immediately before artificial insemination (AI). Heat stress-related factors and reproductive tract scores were accessed using AI. Temp = temperature; RH = relative humidity; THI = temperature humidity index; RBC = red blood cells; Hb = hemoglobin; PCV = packed cell volume; WBC = white blood cells; ALT = alkaline aminotransferase; ALP = alkaline phosphatase; BHB = beta-hydroxybutyric acid; NEFA = non-esterified fatty acid. * All artificial insemination was performed by only one inseminator.

**Figure 2 animals-14-01763-f002:**
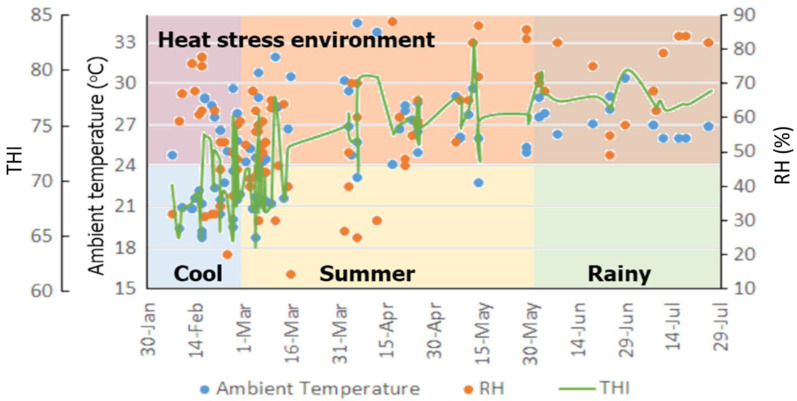
Ambient temperature (°C), relative humidity (RH, %), and temperature humidity index (THI) in relationship with calendar dates and seasons during AI of dairy cattle at 24 h after estrus in a tropical environment (n = 108). Heat-stress environment, defined when THI > 72, showed on the red, orange and brown area.

**Figure 3 animals-14-01763-f003:**
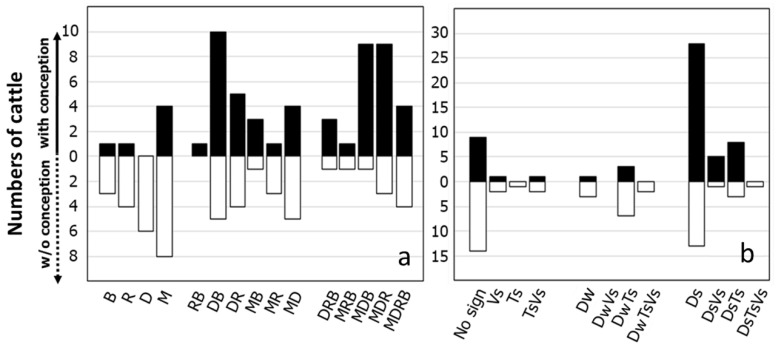
Numbers of cattle showing estrus signs and their combinations (**a**) and reproductive tract characteristics at AI (**b**) and their association with conception outcome. Estrus signs included bellowing (B), restlessness (R), vaginal discharge (D), and mounting (M). Reproductive tract characteristics included cervical discharge that was watery (Dw) or sticky (Ds), vulva swelling (Vs), and strong tone (Ts).

**Table 1 animals-14-01763-t001:** Number and percentages of AI services, animals observed in standing estrus, conception rate, and conception rate by season based on voluntary time from detection of estrus to AI at 12, 24, and 36 h.

		Overall		12 h		24 h		36 h	
		n	%	n	%	n	%	n	%
Number of AI		160		26	16.25	108	67.5	26	16.25
Standing estrus		5	3.1	1	3.9	3	2.8	1	3.9
Conception rate *		79	49.4	10	38.5 ^a^	58	53.7 ^b^	11	42.3 ^a^
Conception rate by season *									
	Cool	34	69.4	7	58.3 ^x^	23	71.9 ^y^	4	80 ^y^
	Summer	68	42.5	0	0	28	51.9	5	45.5
	Rainy	43	26.9	3	27.3	7	31.8	2	20

^a,b^ Means in the same row with different superscripts differ (*p* < 0.05). ^x,y^ Means in the same row with different superscripts tend to differ (*p* < 0.10). * Repeated logistic regression was used to determine the effect of time to AI on conception risk.

**Table 2 animals-14-01763-t002:** Estrus signs and factors measured at AI in association with conception at *p* < 0.2 using univariable analyses of repeated logistic models.

		Odds Ratio	95% Confidence Limits		
Parameter			Lower	Upper	*p-*Value
*Environment and animal factors*					
Parity	Heifer	11.65	0.77	177.10	0.08
	>1	1.73	0.71	4.19	0.23
	1	1.00			
Season	Cool	4.92	1.58	15.33	0.01
	Summer	1.93	0.66	5.65	0.23
	Rainy	1.00			
Body temperature, °C		0.25	0.07	0.83	0.02
Relative humidity, %		0.96	0.93	0.99	0.01
*Estrus expression before artificial insemination*					
Bellowing		2.62	1.21	5.67	0.01
Vaginal discharge		2.52	1.39	4.54	<0.01
Vaginal discharge/bellowing		3.27	1.57	6.81	<0.01
Mounting/bellowing		2.52	0.75	8.39	0.13
Mounting/vaginal discharge		2.28	0.93	5.61	0.07
Mounting/vaginal discharge/bellowing		2.96	0.57	15.32	0.20
Number of observed	3	9.21	3.30	25.71	<0.0001
estrus signs	2	4.56	2.27	9.15	<0.0001
	1	1.00			
*Reproductive tract characteristics at AI*					
Cervical discharge	Sticky	5.97	2.34	15.19	<0.001
	Watery	1.47	0.55	3.96	0.44
	None	1.00			
*Blood chemistry and reproductive hormones*					
Hemoglobin		1.33	1.03	1.72	0.03
Hematocrit		1.09	0.99	1.19	0.08
Alkaline phosphatase *		2.61	0.95	7.17	0.06
Creatinine		4.45	0.52	38.09	0.17
Estradiol *		1.21	1.00	1.46	0.05

* Data were subjected to logarithm transformation.

**Table 3 animals-14-01763-t003:** The final model of estrus signs and factors is measured at AI 24 h after seeing the secondary signs of estrus in association with conception using multivariate analyses of repeated logistic models.

			95% Confidence Limits		
Parameter		Odds Ratio	Lower	Upper	*p*-Value
*Environment and management factors*					
Parity	Heifer	8.58	0.96	76.41	0.05
	>1	2.51	0.36	17.61	0.36
	1	1.00			
Body temperature (°C)		0.19	0.03	1.17	0.07
Relative humidity (%)		0.93	0.87	0.99	0.02
*Estrus expression before artificial insemination*					
Number of observed	3	23.71	2.28	246.93	0.01
estrus signs	2	12.82	4.98	32.97	<0.0001
	1	1.00			
*Reproductive tract characteristics at AI*					
Cervical discharge	Sticky	6.85	2.85	16.47	<0.0001
	None	5.96	1.77	20.04	<0.001
	Watery	1.00			

## Data Availability

The data presented in this study are available upon request from the corresponding author.

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
