# Peer review of "Factors at Insemination and Subsequent Conception of Cattle under Heat-Stress Tie-Stall Environments"

_animals, 2024, doi:10.3390/ani14121763_

Round 1
Reviewer 1 Report
Comments and Suggestions for Authors
I think that this is a well-written paper, and it was easy to follow. There is a good contribution to scientific applied knowledge. Theme is very actual and important, because there are still problems with cow’s reproduction worldwide.
General comments:
The Title, Summary and Abstract cover the content of the manuscript.
The description of the Materials and methods is sufficiently clear and complete. The work has been well designed and interpreted.
The interpretation and discussion of Results is adequate and supported by the data. The conclusions are supported by the results. I have one minor point (recommendation): quality of Figure 2 – that could be presented in better quality.
In my opinion, after carefully studying of this paper, the present manuscript should be accepted for publication in Animals.
Author Response
Thank you very much.
Reviewer 2 Report
Comments and Suggestions for Authors
In the article, the authors examined insemination factors such as insemination time, secondary signs of estrus behavior, reproductive system characteristics and heat stress factors in cattle living in tied-stall barns in a tropical environment. The subject of this study is suitable for the “Animals” journal. The authors argue that for cows housed in tie-stall barns in a tropical environment characterized by heat stress, delaying AI until 24 hours after detecting secondary signs of estrus causes higher pregnancy rates. On the other hand, they found that relative humidity and body temperature on the day of insemination reduced the risk of pregnancy. There are many studies to determine the effects of heat stress on reproductive performance, estrus behavior, AI success rate, plasma chemicals, metabolites, and reproductive hormones in dairy cattle. The current study could not go beyond repeating previous studies. Therefore, it is an article with low originality. For all these reasons, publishing the article in the "Animals" journal is inappropriate.
Author Response
Thank you for the comments. Our heat stress condition is in a tropical environment that is completely different from the heat stress in the temperate zone, where the heat stress occurs during summertime. In Thailand, dairy cattle living in a hot-humid environment are confronted with heat stress almost all the time. Therefore, most dairy cattle could partly adjust themselves to the variety of levels of heat stress. Our findings, either similar or different, are reflected in the specific heat stress in the tropics. We hope that our findings could help animal scientists understand more about the mechanism of heat stress in the tropics.

Reviewer 3 Report
Comments and Suggestions for Authors
The practical interest of the research is high attending the conditions of animals housing. Nevertheless, it as been done with a relatively small number of animals. Unfortunately, any information on follicular development at AI has been collected due to the absence of ultrasound device. Many blood parameters have been evaluated. It seems necessary to justify more their use.
Some more specific comments
11 thermal stress can be also observed in subtropical and temperate regions
37 Could you briefly explain in the introduction the interest of blood and biochemical analysis at the time of insemination ?
45 cows are also concerned
46 how can you explain the possibility to observe head butting, licking, and mounting in tie-stalls ?
48 the relative importance of oestrus signs have been revieuwed by Roelofs Roelofs JB, van Eerdenburg FJCM, Soede NM, Kemp B.Various behavioral signs of estrus and their relationship with time of ovulation in dairy cattle. Theriogenology 2005;63: 1366–77.
74 could you give an idea of te average milk production level ?
77 how many heifers ?
81 Do you think that such lenght of the ropes are sufficient to observe mounting activities ?
87 how many times per day the farmers are looking for oestrus signs ?
95 could you give some more informations on the aveargae value of T° and humidity in the three considerd periods namely cool, summer and rainy
99 can you confirm that all inseminations have been done by only one inseminator ?
109 what kind of devices have been used to measure temperature and humidity in each farm at each insemination ?
140 figure 1 could you explain how mount and standing heat can be observed if the animals are tied ? Moreover how have you evaluated the redness of the vulva ? What the pink vulva membrane ? Uterine tone : what about no tone i.e flaccid uterus ? Could you better define the consistency of the mucus : what really means watery and sticky ?
154 could you justify why you have excluded cows inseminated 12 and 36 h after seconday signs of oestrus detection ? Their results are nevertheless presented in the table 1.
189 two times the same observation : The conception rate of cattle inseminated at 24 h after estrus was higher P < 0 05 than that observed at 12 h after estrus. Only 5 or 3.1% of the cattle included in the study were reported to be in standing estrus The highest conception rate 53 7% observed was for cattle inseminated at 24 h after estrus, resulting in a higher risk of conception than either 12 or 36 h OR 0 39 and 0 44, respectively
191 5 animals have been seen to be mounted : it’s quite normal attending the tie-stalls
216 Figure 2 What’s the interest of the graph ? Moreover we do’nt really see the red watermark area
234 Rutland in 2005 observe : Cervical mucus becomes more plentiful,watery, translucent, less viscous and easier to traverse by spermatozoa in the follicular phase of the ovarian cycle. By contrast, in the luteal phase of the cycle this mucus becomes scanty and viscous and,consequently, unfavorable to sperm penetration (Rutllant J, López-Béjar M, López-Gatius F. Ultrastructural and rheological properties of bovine vaginal fluid and its relation to sperm motility and fertilisation: A review. Reprod Domest Anim 2005;40:79 –96.
238 The sensitivity and sensibility of signs observed before AI and at AI should have been analyzed. Moreover, the evaluation of the follicle diameter has not been done at AI. This sign is very important.
283 The 12-hours delay is right is a mounted activity is detected. Yoy must add that 24 hour of delay is secondary signs like bellow or vaginal discharge or bellow and vaginal discharge or … are detected
306 this sign can be observe dis the ground is not too slippery
350 are you able to propose a cutt-off for body temperature ?
351 the descriptive data of body temperature has not been presented in the results. Which higher temperature has been considered ?
361 what about the results of others hormones like progesterone, LH and FSH? Can you add some comments about this positive correlation between conception rates and haematocrit, phosphatase, creatinine and hemoglobin
Author Response
The authors are appreciated all comments and corrections from reviewers We believe that the revised manuscript is more understandable and valuable for the reader. All corrections were presented in the attached additional file.
